# Endothelial-to-Mesenchymal Transition in an Hereditary Hemorrhagic Telangiectasia-like Pediatric Case of Multiple Pulmonary Arteriovenous Malformations

**DOI:** 10.3390/ijms25116163

**Published:** 2024-06-03

**Authors:** Laura Lorente-Herraiz, Angel M. Cuesta, Lucía Recio-Poveda, Luisa M. Botella, Virginia Albiñana

**Affiliations:** 1Departamento de Biomedicina Molecular, Centro de Investigaciones Biológicas Margarita Salas, CIB-CSIC, 28040 Madrid, Spain; laura.lorente@cib.csic.es (L.L.-H.); lucia.recio@cib.csic.es (L.R.-P.); cibluisa@cib.csic.es (L.M.B.); 2Centro de Investigación Biomédica en Red de Enfermedades Raras (CIBERER), ISCIII, 28029 Madrid, Spain; angcuest@ucm.es; 3Departamento de Bioquímica, Facultad de Farmacia, Universidad Complutense de Madrid, 28040 Madrid, Spain; 4Instituto de Investigación Sanitaria del Hospital Clínico San Carlos (IdISSC), 28040 Madrid, Spain

**Keywords:** rare vascular disease, Hereditary Hemorrhagic Telangiectasia (HHT), pulmonary arteriovenous malformations (PAVMs), endothelial-to-mesenchymal transition (EndMT), angiogenesis, TGF-β

## Abstract

Pulmonary arteriovenous malformations (PAVMs) are vascular anomalies resulting in abnormal connections between pulmonary arteries and veins. In 80% of cases, PAVMs are present from birth, but clinical manifestations are rarely seen in childhood. These congenital malformations are typically associated with Hereditary Hemorrhagic Telangiectasia (HHT), a rare disease that affects 1 in 5000/8000 individuals. HHT disease is frequently caused by mutations in genes involved in the TGF-β pathway. However, approximately 15% of patients do not have a genetic diagnosis and, among the genetically diagnosed, more than 33% do not meet the Curaçao criteria. This makes clinical diagnosis even more challenging in the pediatric age group. Here, we introduce an 8-year-old patient bearing a severe phenotype of multiple diffuse PAVMs caused by an unknown mutation which ended in lung transplantation. Phenotypically, the case under study follows a molecular pattern which is HHT-like. Therefore, molecular- biological and cellular-functional analyses have been performed in primary endothelial cells (ECs) isolated from the explanted lung. The findings revealed a loss of functionality in lung endothelial tissue and a stimulation of endothelial-to-mesenchymal transition. Understanding the molecular basis of this transition could potentially offer new therapeutic strategies to delay lung transplantation in severe cases.

## 1. Introduction

Pulmonary arteriovenous malformations (PAVMs) are the consequence of an incorrect connection between pulmonary veins and arteries, leading to dilated anomalous vessels and the disappearance of the capillary network. This generates a right-to-left shunt, which increases pressure in the pulmonary circulatory system, as well as impairing regular gas exchange [1,2].

PAVMs are typically asymptomatic and undiagnosed. Approximately 80% of PAVMs are congenital in origin, but they are not commonly diagnosed in children because clinical manifestations are rarely seen before adulthood. However, patients who remain undiagnosed may develop serious complications later in life. The severity of the clinical symptoms can vary from dyspnea to cyanosis, heart palpitations and chest pain [3,4].

The majority of patients with congenital PAVMs have underlying Osler–Weber–Rendu syndrome, also known as Hereditary Hemorrhagic Telangiectasia (HHT). HHT is a rare multisystemic vascular disease with an autosomal dominant pattern of inheritance and a prevalence of 1 in 5000/8000 in the general population, which increases in certain regions due to the founder effect and geographical isolation [5,6]. Patients with HHT often present mucocutaneous telangiectasias, the rupture of which causes recurrent and spontaneous nose hemorrhages (epistaxis), which is usually the most frequent and the first symptom to appear in HHT. Bleeding usually increases throughout the patient’s life, and in some of the most severe cases can even lead to anemia and iron deficiency. Another clinical manifestation is the presence of arteriovenous malformations (AVMs) in internal organs, such as the lungs and brain, more frequent in the HHT1 type, while in HHT2 it is more frequent in the liver [7]. Clinical diagnosis of HHT is possible following the Curaçao criteria, which are epistaxis, telangiectasia, arteriovenous malformation in internal organs and a familiar pattern of autosomal dominant inheritance [8]. Nevertheless, the Curaçao criteria are not accurate for infants, due to the late development of certain symptoms and to the intra-familial phenotypic variability [9].

In addition to a symptomatic perspective, HHT can be diagnosed genetically. All mutations leading to HHT are found in genes belonging to the BMP9/TGF-β signaling pathway. Mutations in the *ENG* (HHT1) and *ALK1/ACRVRL1* (HHT2) genes are responsible for 90% of HHT cases [10,11]. Less common mutations, such as those found in *MADH4/SMAD4* and *GDF2* genes, have been described in fewer than 2% of the HHT population and result in HHT-like syndromes. Mutations in *MADH4/SMAD4* cause juvenile polyposis/HHT overlap syndrome (JPHT), which generates, additionally to HHT symptoms, colon polyps and thoracic aneurysms [12]. In the case of *GDF2* mutations, they generate an HHT-like syndrome classified as HHT5 [13]. Recent research suggests that mutations in *EPHB4* or *RASA1* may be responsible for those previously known as HHT3 and HHT4 types, as there is no evidence for the existence of two independent genes linked to chromosomes 5 and 7 [14]. *EPHB4* and *RASA1* variants cause capillary malformation–arteriovenous malformation (CM)/AVM syndrome. This syndrome is characterized by the appearance of multiple small, red, randomly distributed spots with a white halo known as capillary malformations. It can be difficult to distinguish CM/AVM from HHT [15].

The *BMPR2* gene is also involved in the BMP9/TGF-β signaling pathway. Its expression is particularly high in pulmonary vascular endothelium. Pathological changes in this gene can lead to hereditary pulmonary arterial hypertension (hPAH) [16,17]. Moreover, several descriptions of patients with both PAVMs and hPAH with pathogenic BMPR2 mutations have been also described [18,19].

However, 15% of clinically diagnosed subjects with HHT lack a genetic diagnosis. Furthermore, we can add the difficult diagnosis in pediatric cases and the existence of sporadic (non-inherited) cases of HHT. In this context, we have studied a clinical case with multiple diffuse pulmonary fistulas of pediatric age who underwent a bilateral lung transplantation [9]. Previous genetic analysis such as Sanger sequencing analysis and Optical genome did not show any genetic alterations associated with HHT or any of the other related syndromes mentioned above. Moreover, the patient does not have any family history of HHT. Given the severity of the clinical manifestations and their similarity to HHT, the aim of the present study is to establish the molecular parameters and differential gene expression profile of this symptomatology, aiming to establish a similarity to HHT.

## 2. Results

The Index case of this study is an 8-year-old child with clinical diagnosis of probable HHT without a family history of HHT. The case shows incomplete Curaçao criteria: epistaxis and multiple diffuse PAVMs in both lungs. The symptoms had been present since he was 3 years old, including strong cyanosis, and hypoxemia (abnormal low oxygen in blood). This phenotype was becoming more severe and non-compatible with life, due to the low oxygen saturation. In 2020, he was put at the top of an emergency list for pediatric lung transplantation. In spite of all the genetic analysis carried out in the known genes related to HHT and HHT-like syndromes, the genetic origin of the disease remains unknown. Sanger sequencing analysis of both germ-line DNA (peripheral blood) and from DNA of lung tissue was performed. The analysis included the exon and intron-exon boundaries of the two most common HHT pathogenic genes: *ENG* and *ACVRL1/ALK1*. After obtaining a negative result, we carried out a genomic analysis of the classical and other HHT-related genes with specific capture of *ENG*, *ACVRL1/ALK1*, *MADH4/SMAD4*, *BMP9* and *RASA1* chromosome regions (Illumina platform). The analysis was carried out in DNA from the blood and lung. To discard big genomic reorganizations, we performed an optical genome of DNA samples from the blood and lung. Clinical exome sequencing was also carried out in germline DNA of the test sample and in samples from his mother and his sister as control. His father had died years ago from an accident. Clinical exome sequencing with a more than 500-read depth, was carried out with the lung DNA to discard possible mosaicism.

### 2.1. Visual Analysis of Primary Endothelial Cell Culture

Figure 1 displays photographs of two cultures of human pulmonary microvasculature: the Index case and the HMVEC_L as control of lung microvasculature. As a routine way to confirm the identity of our established endothelial cell culture, we performed CD31/PECAM-1 and vWF staining. The cultures appear similar at first, but over time the cells from the case study changed in size and morphology. In the primary culture of the patient, two cell populations can be identified. One population maintains an endothelial appearance, while the other acquires a mesenchymal-like morphology, and a clear increase in size is apparent. Ultimately, the mesenchymal-like morphology became the dominant population in the culture.

Overall, the appearance of this mesenchymal-like cell population over time suggests the presence of an endothelial–mesenchymal transition (EndMT). Due to the prematurity and severity of the symptoms in this child, as well as the observed phenotypic in in vitro changes in cells, we investigated the possibility of molecular-level alterations. As a result, experiments were designed to test the EndMT hypothesis by analyzing the expression and functional profile of the ECs from the explanted lung versus the control HMVEC_L.

### 2.2. mRNA Expression Profile

In the present case, in which there was a vascular involvement, we decided to focus our gene expression analysis on the expression of endothelial and mesenchymal genes. The amount of RNA was normalized to the amount of 18S RNA as a housekeeping gene. Figure 2 provides additional details about the mRNA profile.

The gene expression was compared between the primary cell culture of ECs from a healthy donor and the Index case. RT-qPCR analysis showed that all of the endothelial markers were downregulated in the case under study (Figure 2A). Although these results are to be expected in HHT, as seen in the literature [20], the expression levels were extremely low. In HHT patients, endoglin haploinsufficiency leads to increased COX2 expression and decreased NO synthesis, which is a compensatory mechanism [21] However, in this case, the compensatory system is not fulfilled (Figure 2B). On the other hand, the Index case expressed high mRNA levels of mesenchymal markers, such as *MMP3*, *FN1* and *Vimentin* (Figure 2C). We did not find significant differences in *SNAI1*, whose functions are associated with developmental processes, neural differentiations and epithelium–mesenchymal transition [22]. The replacement of usual epithelial markers with mesenchymal markers, such as Vimentin and Fibronectin, determines these changes [22,23]. Figure 2A,C confirm a decrease in endothelial markers and an increase in mesenchymal markers, supporting our initial idea that an EndMT is occurring. EndMT happens during embryonic development, but this process has been also previously described in adults [24,25].

Also, we examined the expression of *SMAD* genes, since they play a significant role in the TGF-β pathway affected by the HHT syndrome. A downregulation of *SMAD* genes is observed (Figure 2D).

### 2.3. Protein Expression

To corroborate the previous expression profile comparing HMVEC_L with the Index case, we performed protein analysis by immunofluorescent assay and/or Western blot.

An immunofluorescent assay was conducted to corroborate the gene expression levels of endothelial cell-specific genes (VE-Cadherin 2, PECAM-1) and a specific marker of dedifferentiated cells (N-Cadherin) at the cellular level (Figure 3A). The images confirm the presence of VE-Cadherin 2 on the cell surface of HMVEC_L control culture cells, observing strong accumulations of this protein observed in the areas of cell–cell contacts, forming adherent junctions. In contrast, VE-Cadherin 2 is less prominent in the patient’s cells, and there is no accumulation of it on the cell surface. This suggests that cell–cell contacts have been lost. Since VE-Cadherin 2 plays a crucial role in maintaining endothelial integrity and leukocyte extravasation, a decrease in VE-Cadherin 2 would result in reduced cell adhesion and less cohesion among cells. To confirm the endothelial identity of the cell culture, CD31/PECAM-1 labelling was used as marker. This protein is almost exclusively expressed in ECs. The patient cells exhibited significantly lower levels of PECAM-1 compared to the control cells, which supports the results obtained from qPCR and confirms our initial hypothesis.

Furthermore, an examination of F-Actin in both cultures was conducted to determine whether the mesenchymal-shaped cells showed a different cytoskeleton distribution from that of healthy ECs, which might facilitate proper migration and contraction. Figure 3B illustrates significant disparities in the quantity and configuration/organization of actin filaments. When comparing both conditions, it is evident that the labelling of the control cells enables a precise delimitation of the cell shape, which is not the case in the patient samples. In the latter, the labelling accumulates in cytoplasmic fibers, which are thicker and more abundant than in the control. Additionally, a significant variability in cell size within the same culture was observed during the experiment (see Figure 3B). The larger, apparently mesenchymal, cells showed a highly ordered network of filaments. The phase contrast allowed for the identification of smaller cells as ECs undergoing apoptosis (Figure 3C).

To analyze TGF-β and PI3K/MAPK pathways, we performed Western blot assays. The TGF-β pathway is closely related to the HHT syndrome and plays a crucial role in a wide range of cellular functions. The signaling status of the TGF-β pathway was analyzed through SMAD proteins (SMAD1, SMAD3 and SMAD4). We examined the levels of each of the SMAD proteins and their response to TGF-β treatment (Figure 4A). Figure 4A shows that the Index case has a complete inhibition of this pathway, and TGF-β stimulation has no effect. The near absence of SMAD4 expression in the patient’s endothelial cells suggests the notion that SMAD4 is a central mediator of ENG/ACVRL1/ALK1 signaling, indispensable for the development of a proper arteriovenous network. In this case, a lack of SMAD4 in the pulmonary ECs, plus extremely low levels of SMAD1/3, ends in a completely altered pulmonary arteriovenous network [26].

Considering the immunofluorescence results, the use of β-Actin as a loading reference for the patient was not optimal. However, the absence of SMADs in the patient was so evident that the loading control was dispensable.

Additionally, we aimed to investigate the status of the PI3K/MAPK pathway and its activation (phosphorylated proteins). Figure 4B illustrates that all proteins from the Index case were overexpressed when compared with the control, the healthy HMVEC-L cells, indicating a higher activation of both pathways through AKT/pAKT and ERK/pERK. Additionally, we aimed to examine the protein levels of the mesenchymal marker Vimentin. As previously observed in the RT-qPCR, the Western blot assay revealed a higher quantity of Vimentin in the Index case, while it was undetectable in the control (Figure 4B).

### 2.4. Functional Analysis

Due to the low expression levels of certain genes involved in processes such as angiogenesis (*ANGPT2*), migration, and cell adhesion (*PECAM-1* and *VE-Cadherin 2*) and genes involved in normal blood vessel physiology (*NOS-3* and *COX2*), we performed functional assays of wound healing and tube formation (angiogenesis) to test how compromised patient cells were compared to healthy control cells.

Figure 5A shows the wound healing assay. These assays attempted to mimic a physiological condition of in vitro tissue damage and test cell function by assessing cell migration and cohesion. In both cases, the advancing front moved. However, it is evident that HMVEC_L control cells exhibited a more organized and uniform movement, while the cells of the patient under study displayed less connection, with each one moving independently. It is obvious that the control cells showed a greater healing capacity, as they were capable of closing almost completely the discontinuity (wound) after 6 h. In contrast, the cells in the case under study moved towards the wound in an attempt to close it, but after 24 h gaps between the cells can be observed and the closure is shown to be incomplete. Therefore, the control cells have a greater capacity for repairing and forming cell cohesion, while the patient cultures experience delays and incomplete closure.

In terms of functional capacity for tubulogenesis, Figure 5B illustrates the fact that healthy HMVEC_L cells were able to generate completely closed and defined tubes of various sizes within 3 h on Matrigel, BD (Franklin Lakes, NJ, USA)^®^. After 6 h, the tubes formed were reinforced by several layers of cells. However, cells from the Index case were unable to build defined tubes, and a large proportion of the cells were not able to adhere to Matrigel^®^ after 6 h of plating.

## 3. Discussion

Angiogenesis processes play a dual role: they are essential for the formation of new vessels from pre-existing ones but, when impaired, are also involved in pathogenic processes. ECs undergo morphological changes and are induced to migrate during angiogenesis, which enables them to migrate and remodel the ECM. This behavior may resemble that of mesenchymal cells. However, it is not typical for ECs to detach from their neighbor cells, indicating that angiogenesis may involve a partial transition to mesenchymal cells, EndMT, followed by vessel stabilization [27,28]. EndMT has also been observed in processes of cardiac fibrosis and pulmonary hypertension [24,29].

This paper presents the first documented case of endothelial-to-mesenchymal transition in a sample from a pediatric patient, subjected to lung transplantation. Although this transition has been observed in adults [24,30], this may be the only documented case in children, so far. Our hypothesis is that damaged pulmonary ECs eventually transition into mesenchymal cells as a survival mechanism when they can no longer fulfil their endothelial function. In spite of the unknown origin of this damage, which is currently under investigation, we have studied the outcomes. The consequences are the loss of endothelial integrity, which could explain the severity of the patient’s symptoms, due to a collapse of the lung vasculature and therefore the lung function. The results of the in vitro experiments described in this work support this hypothesis.

The primary lung microvascular culture from the Index case showed the progressive emergence with passages of a second population with a mesenchymal-like morphology (Figure 1). To test this assumption, we compared the expression profile of this cell culture with a primary control culture (Figure 2). The analyzed genes are involved in various physiological processes, including angiogenesis, cell adhesion, migration, transmigration, TGF-β signaling, cell cycle control, cytoskeleton, and vascular physiology. A decrease in the expression of vascular genes was observed, as has been described in HHT patients. However, the characteristic genes of mesenchymal cells involved in transition processes, such as *MMP3*, *FN1*, *Vimentin* and *SNAI1*, were found to be significantly increased in their expression, in support of our EndMT hypothesis [28,31].

During EndMT, ECs detach and migrate, potentially invading the underlying tissue. As part of this process, the mesenchymal phenotype overlaps with the endothelial phenotype. This is characterized by the acquisition of mesenchymal markers, such as N-Cadherin, and the complementary loss of endothelial markers, such as CD31/PECAM-1 and VE-Cadherin 2. Additionally, ECs lose their cell–cell junctions and acquire characteristics adapted to migration and invasion [32].

To verify these changes, the expression and subcellular localization of these proteins was examined using laser confocal microscopy (Figure 3B). Our findings confirmed that the cells in the case study had lost the cell–cell connections typical of endothelial tissue and were expressing mesenchymal markers (N-Cadherin). Significant oscillations in VE-Cadherin 2 levels can increase N-Cadherin expression in the endothelium. N-Cadherin promotes cell migration and contributes to vascular elongation, while VE-Cadherin 2 induces vascular stabilization by limiting growth factor receptor signaling [33].

Several studies have shown that the TGF-β pathway promotes EndMT [30,34]. TGF-β can also activate SMAD-independent pathways such as MAPK/ERK/JNK, all of which are involved in EndMT [35]. This is similar to the situation we observe in the Index patient, where a total inhibition of the SMAD protein-dependent TGF- β pathway is found [36]. On the other hand, both in HHT patients and in our case an increase in basal and functional protein levels of the PI3K/MAPK pathways is found [37,38]. These results support the hypothesis of an EndMT being present in the case under investigation in this work.

Recent studies suggest that vascular lesions in HHT may be originated by a second mutation event in *ENG* or *ALK1*, in angiogenic processes [39]. Mouse models of HHT1 and HHT2 provide strong evidence supporting a two-hit mechanism as the origin of AVM. This mechanism requires complete endothelial loss of the ENG or ALK1 protein for the formation of robust vascular lesions. The current heterozygous mouse model, with only one alteration, is not effective in replicating HHT symptoms [40]. Singh et al. (2020) [41] noted that AVMs are primarily of venous origin, where cells express higher levels of endoglin. ENG restricts endothelial proliferation, and its absence in localized regions increases the frequency of AVMs [41]. Endoglin is also involved in cell intercalation processes, necessary for the extension and formation of new blood vessels while maintaining cell–cell contact. Therefore, its absence may be a factor in the development of these vascular lesions [42].

HHT patients exhibit at least a 50% reduction in ENG or ALK1 functional protein levels, due to their heterozygous condition for mutations in *ENG* (HHT1) or *ALK1* (HHT2). These patients experience increased and anomalous angiogenic processes, where there is no proper control on the generation and stabilization of blood vessels. The reported finding suggests that low ENG levels adversely affect both cell migration and tubule formation [43,44,45]. Previous studies in human brain AVMs have found evidence that while SMAD-mediated TGF-β signaling is reduced in AVMs, non-canonical TGF-β signaling may contribute to the development of AVMs through the PI3K/MAPK pathway [36,46]. In HHT, due to mutations in components of the TGF-β pathway, a decrease in the canonical TGF-β pathway and an increase in the PI3K pathway is found, facilitating a partial endothelial–mesenchymal transition. In this scenario, some cells may undergo a partial EndMT transition in the AVM. However, cells are able to reverse, at least to some extent, the EndMT transition, and HHT cells still maintain cell–cell junctions to an intermediate degree between the healthy cells and the damaged cells observed in the present case [20].

Moreover, in low levels of physiological oxygen saturation such as those exhibited by the patient under study, 70–80% is sensed by cells, as hypoxia, inducing an increase in VEGF levels, and leading to a continuous pro-angiogenic state. Additionally, we observed a very strong decrease, far below 50%, in the expression of both ENG and ALK1, close to the conditional knock-out mouse models of *Eng* and *Acvrl1/Alk1* published in the literature [42]. The study found a phenotype with greater aggressiveness in the knockout mouse of *Acvrl1/Alk1*, resulting in a complete inhibition of the TGF-β pathway, as seen in our Index case (Figure 4A) [40]. Additionally, decreased levels of ENG create an adequate environment for the generation of multiple AVMs, in our case, pulmonary. A decrease in the levels of ALK1 and ENG impairs TGF-β signaling; addiction to a hypoxic environment leads to an EndMT and finally to the generation of AVM. In the Index case, the ECs undergo the EndMT, becoming individual cells without a proper alignment with other cells, apparently unable to reverse to the endothelial cell phenotype. This results in a permanent EndMT that completely compromises the integrity and function of the vasculature. Furthermore, Figure 3C demonstrate a significant variation in cell size, with some endothelial-like cells undergoing apoptotic processes. This suggests that ECs that are incapable of initiating an EndMT transition process may ultimately die (Figure 5B,C). We identified similarities in gene expression patterns and in the signaling related to TGF-β and PI3K/MAPK pathways between HHT patients and our case study [20,37,38]. However, this case presents an extreme profile with low expression of endothelial markers and increased expression of mesenchymal cell-specific markers. This may explain a fatal compromise of vascular integrity and function.

The mechanism explained above is depicted in Figure 6, where three angiogenic situations are shown. The normal angiogenesis with transitory and reversible EndMT (Figure 6) is undergone by healthy ECs. The transition with only partial reversion would occur in an HHT patient. Finally, the EndMT suffered by the cells presented in this case under study would lead to a loss of endothelial function in the vasculature. Although the origin of the anomalous transition which occurred in this pediatric patient remains unknown, it is likely that this could have occurred somatically, restricted mainly to the lung, without involvement of the germline.

## 4. Materials and Methods

### 4.1. Human Samples and Ethics

The experiments included in this work were approved by the Ethical Committee of CSIC with the reference Number 075/2017. All donors were informed of the experimental procedures, and their informed consent was obtained. The guardian of the patient studied in this manuscript gave his informed consent for the patient to participate in the study.

Cells derived from patient’s explanted lung were isolated and cultured.

### 4.2. Cell Culture

Primary cultures of lung microvasculature cells from the patient under study were isolated following an established procedure from one of the explanted lungs after surgery performed for replacement of both lungs at the Hospital de La Fe, Valencia (Spain). To isolate lung endothelial cells, mechanical deep disruption was made using scalpels to reduce pieces of around 30 mg to a homogenous mixture. EGM-2/EBM-2 basal medium (3 mL) (PromoCell, Heidelberg, Germany) without serum was added to the lung mixture and subjected to collagenase digestion for 20 min at 37 °C with gentle shaking in 5 mL sterile tubes. Then, 20 mL of EGM-2/EBM-2 basal medium (PromoCell) was added, and the mixture was filtered through a 70 µm cell strainer. The strainer was washed with an additional 20 mL of basal medium, and the flow-through was centrifuged at 1250 rpm for 5 min. The supernatant was removed and the pellet was washed with 20 mL Hanks Buffered Solution (HBSS) and centrifuged for an additional 5 min. The resulting pellet was resuspended in 2 mL of EGM-2/EBM-2 endothelial-growth complete medium containing all the specific endothelial supplements (PromoCell), and 20% FBS plus anti mycotic-antibiotic solution (Gibco, New York, NY, USA) and Mycozap as anti-mycoplasm. Cells were plated into 6-well plates previously coated with collagen type I. In the first week of culture, the medium was changed daily and replaced with fresh EGM-2/EBM-2 complete medium (PromoCell). During the first days, many cells originating from the blood were visible on the plates. However, after 10–15 days of culture, clusters of cobblestone-shaped cells became visible. The EGM-2/EBM-2 complete medium (PromoCell) was replaced every day, from the 15th to the 30th day of culture, and no other type was visible in the 6-well plates, except the cobblestone-shaped cells, which became confluent 30 days from the plating day. Cells from each confluent 6-well plate were detached by trypsin, passed to F-25 collagen-coated flasks and cultured for 3 days with 5 mL of EGM-2/EBM-2 complete medium (PromoCell) (P-1 passage) until confluence. Some of the cells were frozen at this stage, (P-1 passage) at a density of around 1 million cells. Some P-1 flasks were expanded further to 5 F-25 mL each, and frozen when flasks became confluent at passage 2 (1 million cells each). A human lung microvasculature endothelial primary-cell culture (HMVEC_L) was used as control (Innoprot, Derio, Spain). Both the patient’s and HMVEC_L cultures required precoated collagen plates (Sigma-Aldrich, St. Louis, MO, USA) for attachment and were grown in EGM-2/EBM-2 medium (PromoCell) supplemented with 20% FBS (Gibco), 2 mM L-glutamine, and 100 U/mL penicillin/streptomycin (Gibco). Experiments were carried out at cell passages lower than 7, and each parallel experiment between control HMVEC_L and patient-derived lung ECs was performed at similar passages. For passaging, cells were detached from the plates with 0.05% Trypsin and 0.02% EDTA solution (Gibco). HMVEC_L and patient-derived lung endothelial cell cultures were treated with two different concentrations of TGF-β (R&D Systems, Miami, FL, USA) to stimulate the TGF-β pathways, either preferentially mediated by ALK5 (treatment with 10 ng/mL o/n), or preferentially by ALK1 (1 ng/mL for 3 h). The treatment was conducted in EGM-2/EBM-2 medium (Promo Cell) with 2% FBS (Gibco), and a control was maintained without any treatment (EGM-2/EBM-2 medium with 2% FBS).

All cell cultures were grown in conditions of 37 °C and 5% CO_2_ and humidity. Cell-culture brightfield images were taken using a Zeiss Axiovert 135 microscope (Oberkochen, Germany).

### 4.3. Gene Expression Analysis at the RNA Level

#### Real-Time Quantitative PCR

Total RNA was extracted from the Index case and control primary cultures using the NucleoSpin RNA kit (Macherey-Nagel, GmbH&Co., Düren, Germany). A total of 500 ng of total RNA from each sample was retrotranscribed into cDNA in a final volume of 20 μL with a commercial High-Capacity cDNA Reverse Transcription kit (Thermo Fisher Scientific, Waltham, MA, USA). This kit uses hexamer random sequencing primers (random primers). The iQ5 system (BioRad, Hercules, CA, USA) was used to perform real-time PCR using FastStart Essential DNA Green Master (Roche, Basel, Switzerland) as reaction mix. The primers used in q-PCR (Sigma-Aldrich) are shown in the Table 1. Each gene was analyzed in triplicate for each of the samples used.

### 4.4. Analysis of Gene Expression at the Protein Level

#### 4.4.1. Immunofluorescence Assay

Immunofluorescence analyses were conducted to test some of the real-time quantitative PCR results and to assess the expression of endothelial cell-specific markers, VE-Cadherin 2 and PECAM-1, as well as the mesenchymal cell marker N-Cadherin. Additionally, we aimed to analyze differences in the cytoskeleton’s F-Actin between control and patient cells.

Cells from both cultures, HMVEC_L and Index case, were plated on pre-coated collagen sterile glass coverslips (12 mm diameter, VWR international, Radnor, PA, USA). The control culture HMVEC_L was seeded with 2 × 10^4^ cells/well, while the patient culture was seeded with 4 × 10^4^ cells/well, due to the special characteristics of cells, as explained in the Results section.

On the following day, the cells were washed with cold 1× HBSS (Gibco) and fixed with 3.5% paraformaldehyde in 1× PBS for 20 min at 4 °C. In the case of F-Actin, we permeabilized with 100 μg/mL L-α-Lysophosphatidylcholine (LPC) (Sigma-Aldrich). The cells were then washed again with 1× HBSS and non-specific unions were blocked with 1% PBS 1×/BSA for 1 h at 4 °C. Mouse anti-human antibodies were used to detect VE-Cadherin 2 (1:200 in PBS 1× BSA 4%) (Abcam, Cambridge, UK), N-Cadherin, and PECAM-1 (1:50 in PBS 1×/BSA 4%) (Santa Cruz Biotechnology, Dallas, TX, USA) at 4 °C for 40 min. Then, cells were washed with 1× PBS and incubated for 30 min at 4 °C with secondary antibodies such as AlexaFluor 488 for VE-Cadherin 2 and N-Cadherin (1:250) and AlexaFluor 647 (BioLegend, San Diego, CA, USA) or AlexaFluor 568 (Molecular Probes, Eugene, OR, USA), for PECAM-1and F-Actin, respectively.

Finally, ProLong mounting medium with DAPI (Invitrogen, Thermo Fisher Scientific, Waltham, MA, USA) was used to mount the cells on glass coverslips. Immunofluorescence images were acquired using an SP5 fluorescence confocal microscope (DMI6000 CS Leica Microsystems, Wetzlar, Germany) at 63× magnification.

#### 4.4.2. Western Blot

Cell lysates were obtained from primary cultures using TNE buffer (50 mM Tris, 150 mM NaCl, 1 mM EDTA and 0.5% Triton X100) and kept on ice for 30 min. Specific protease and proteasome inhibitors (Roche) were added, along with lactocystin (Sigma-Aldrich). The lysates were centrifuged at 14,000× *g* for 5 min. Similar amounts of proteins from the lysates were boiled in SDS sample buffer and analyzed by 4–20% SDS-PAGE gel under reducing conditions (BioRad). The proteins present in the gels were transferred to nitrocellulose membranes (Amersham, Little Chalfont, UK) through electrotransfer o/n. To prevent nonspecific binding, the membrane was blocked with 5% BSA-TBS-0.05% Tween for 1 h, followed by immunodetection with anti-SMAD1/4 (Rabbit mAb; Cell Signaling, Danvers, MA, USA), anti-SMAD3 (Rabbit mAb; Cell Signaling), anti-β Actin (Mouse mAb; Sigma-Aldrich), anti-AKT (Rabbit mAb; Cell Signaling), anti-pAKT (Rabbit mAb; Cell Signaling), anti-ERK (Mouse mAb; Cell Signaling), anti-pERK (Rabbit mAb; Cell Signaling), anti-Vimentin (Mouse mAb, Proteintech, Rosemont, USA) or anti-γ Tubulin (Mouse mAb; Sigma-Aldrich) antibodies. The primary antibody was incubated at 4 °C o/n. Then, samples were washed and incubated with their corresponding secondary antibody from Dako (Glostrup, Denmark) at RT for 1 h. The antibodies were used at the dilution recommended by the manufacturer, and the membranes were revealed using chemiluminescence (SuperSignal West Pico chemiluminescent substrate, Thermo Scientific). FIJI-Image J 1.53q software tool was used to quantify chemiluminescence.

### 4.5. Functional Analysis

#### 4.5.1. Wound Healing

Wound formation assays were generated from scraping confluent cultures of HMVEC_L and the Index case. Both conditions were seeded the day before at a concentration of 8 × 10^4^ cells/well in a 24-well plate. Cells were washed with HBSS (Gibco) and cultured in EBM-2/EGM-2 medium (PromoCell). Assays were monitored and photographed for 6 h with intervals of 90 min in those wells with similarly sized wounds.

#### 4.5.2. Tubulogenesis: Endothelial Cell Tube-Formation Assay

For the angiogenesis or tube formation assay, 8 × 10^4^ cells/well were seeded in 24-well plates. The wells used were previously coated with Matrigel^®^ (Gibco) at a 1:1 dilution in RPMI medium (Gibco), without any supplement. Cultures were monitored and photographed from 90 min after seeding until 6 h; at this time the formed tubes started to retract and finally collapse.

### 4.6. Statistics

Data shown are means ± SD. Those differences shown were analyzed using Student’s *t*-test. Thus, those *p*-values < 0.05 were considered statistically significant and, therefore, in each figure the significance has been marked with asterisks. Values associated with asterisks are * *p* < 0.05; ** *p* < 0.01; *** *p* < 0.001 and **** *p* < 0.0001.

## Figures and Tables

**Figure 1 ijms-25-06163-f001:**
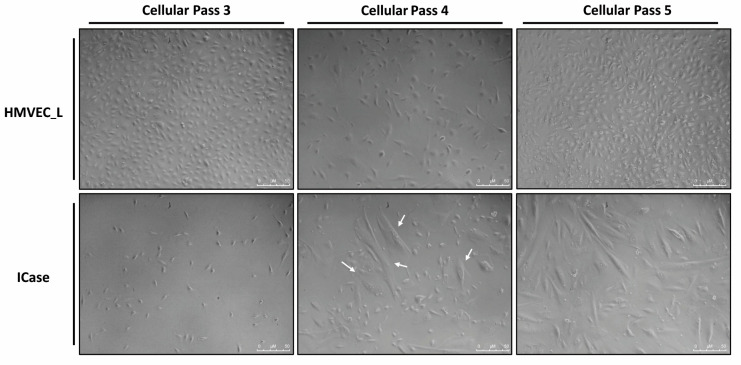
Primary cultures of lung endothelial cells are shown from a control condition (HMVEC_L) and the patient under study (ICase) along different cell passages. The white arrows indicate the appearance of a second, larger cell population, which becomes dominant in the cellular pass 5.

**Figure 2 ijms-25-06163-f002:**
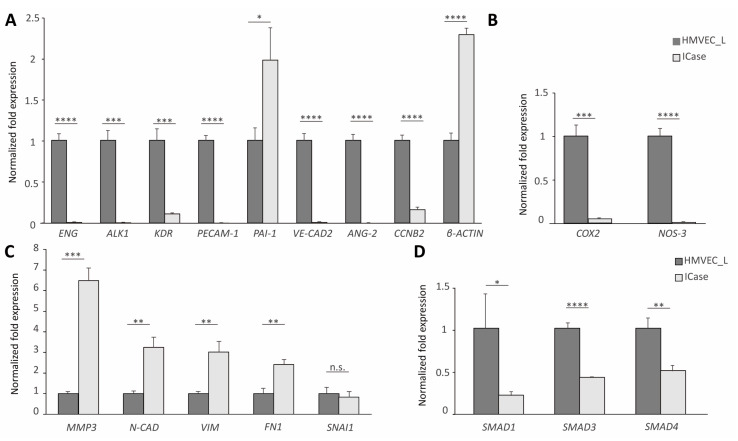
RT-qPCR results for the control (dark gray) and the Index case (light gray). BioRad’s iQ5 Optical System software was used to obtain these values, with the 18S gene used as a housekeeping gene to normalize the results. The β-Actin levels were altered and therefore could not be used as a housekeeping gene (**A**). Differential expression of genes involved in endothelial functions (**A**,**B**), mesenchymal identity, or involved in mesenchymal transition (**C**), as well as genes of the canonical TGF-β signaling pathway, were observed (**D**). Error bars denote ± SEM. Student’s *t*-test. Statistical differences are marked with asterisks: * *p* < 0.05; ** *p* < 0.01; *** *p* < 0.001; **** *p* < 0.0001.

**Figure 3 ijms-25-06163-f003:**
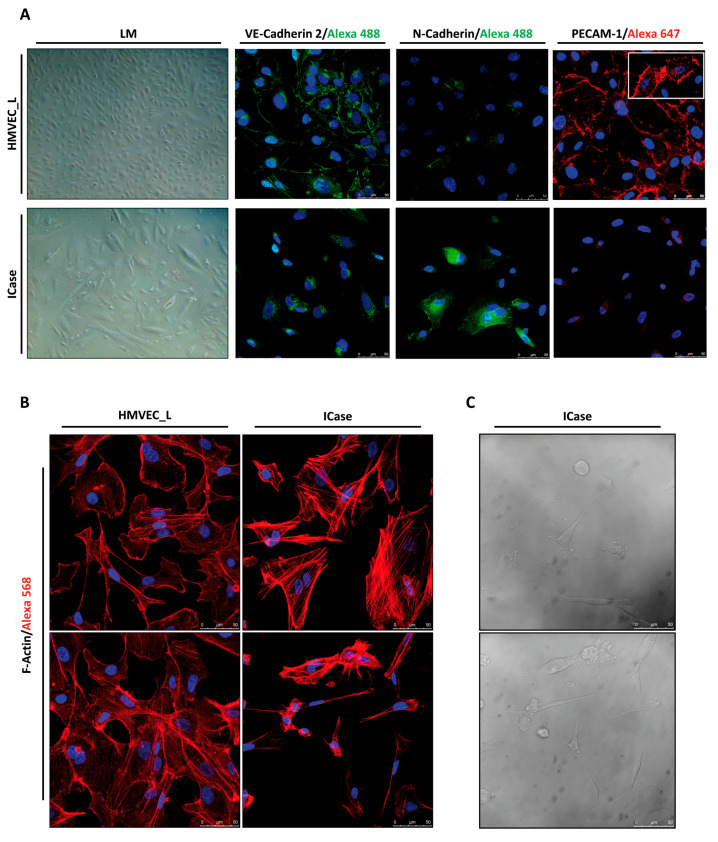
Immunofluorescence, contrast phase, and light-microscopy representative images. The confocal staining of HMVEC_L and ICase displays the distribution of the endothelial markers VE-Cadherin 2 (Alexa 488/green) and PECAM-1 (Alexa 647/red), as well as the mesenchymal marker N-Cadherin (Alexa 488/green) (**A**). In (**A**), the PECAM-1 staining in the upper-left corner captures the moment when filopodia are generated between two cells. Additionally, we used a cytoskeleton marker, F-Actin (Alexa 568/red) (**B**). Nuclei are stained in blue (DAPI). (**C**) displays two fields of the cell culture in phase contrast for the case study (ICase).

**Figure 4 ijms-25-06163-f004:**
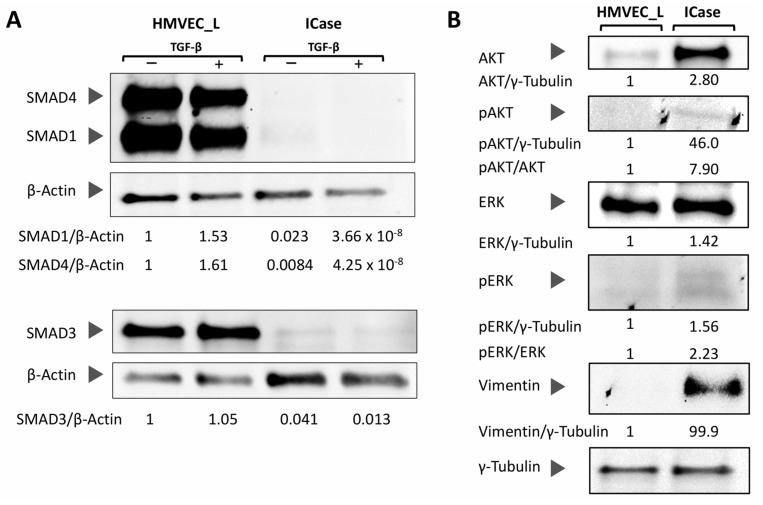
The Western-blot analysis of SMAD proteins involved in signaling through the canonical TGF-β pathway was conducted in control (HMVEC_L) and case study (ICase) cultures, normalized against β-Actin, and treated with TGF-β (+) or not (−) (**A**). (**B**) displays the activation status of the PI3K/MAPK pathway, as well as the levels of the mesenchymal marker Vimentin in both conditions (control and ICase). Additionally, γ tubulin was used as a loading control for all membranes in (**B**). Quantification of the band intensity is shown with the respective load-normalized ratios.

**Figure 5 ijms-25-06163-f005:**
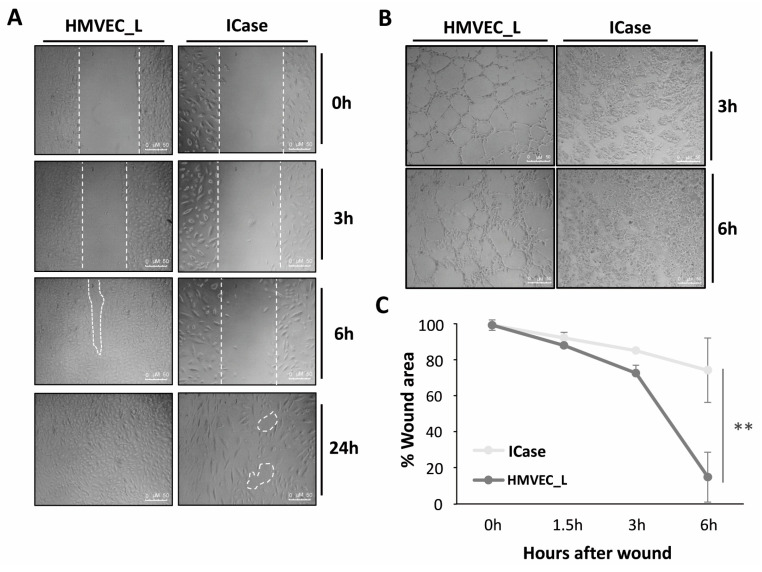
Functional assays in HMVEC_L from a healthy donor (control) and the Index patient (ICase). Wound healing (**A**). A wound was creating by a disruption of the confluent monolayer in both cultures. We monitored cell migration through photos taken at different times. Those areas signaled by white dashed lines mean empty spaces not closed by the patient’s cells during the migration in the wound-healing process, after 24h. Significant differences in the ability to close the wound were observed between the two cultures (**C**). The tube formation process is abnormal or non-existent in the ICase as compared to the control culture (**B**). Error bars denote ± SEM. Student’s *t*-test. Statistical differences are marked with asterisks: ** *p* < 0.01.

**Figure 6 ijms-25-06163-f006:**
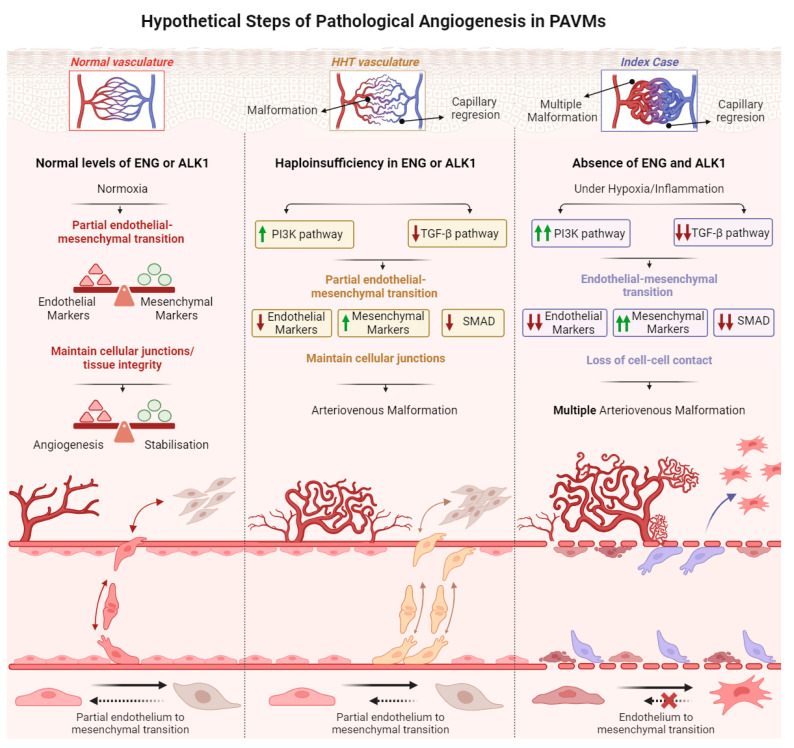
Hypothetical graphic summary of the abnormal angiogenesis in PAVMs, where we observe three states, from the normal to the most severe one.

**Table 1 ijms-25-06163-t001:** Primers used for q-PCR assays.

Genes	Forward Sequences (5′-3′)	Reverse Sequences (5′-3′)
** *18S* **	CTCAACACGGGAAACCTCAC	CGCTCCACCAACTAAGAACG
** *ACVRL1/ALK1* **	ATCTGAGCAGGGCGACAC	ACTCCCTGTGGTGCAGTCA
** *ANG-2* **	TGCAAATGTTCACAAATGCTAA	AAGTTGGAAGGACCACATGC
** *β-ACTIN* **	AGCCTCGCCTTTGCCGA	CTGGTGCCTGGGGCG
** *CCNB2* **	TGGAAAAGTTGGCTCCAAAG	CTTCCTTCATGGAGACATCCTC
** *COX2* **	TCACGCATCAGTTTTTCAAGA	TCACCGTAAATATGATTTAAGTCCAC
** *ENG* **	AGCCACATCGCTCAGACAC	GCCAATACGACCAAATCC
** *FN1* **	TCCATTACCAAGACACACACACT	GGGAGAATAAGCTGTACCATCG
** *KDR* **	GAGTGAGGAAGGAGGACGAAGG	CCGTAGGATGATGACAAGAAGTAGC
** *MMP3* **	CACTCACAGACCTGACTCGGTT	AAGCAGGATCACAGTTGGCTGG
** *N-CADHERIN* **	CCTCCAGAGTTTACTGCCATGAC	GTAGGATCTCCGCCACTGATTC
** *NOS-3* **	GACCCTCACCGCTACAACAT	CCGGGTATCCAGGTCCAT
** *PAI-1* **	TCCAGCAGCTGAATTCCTG	GCTGGAGACATCTGCATCCT
** *PECAM-1* **	AGAAAACCACTGCAGAGTACCAG	GGCCTCTTTCTTGTCCAGTGT
** *SMAD1* **	TTGATGTGCTTTGTGTGCCC	CCACAGCAAAATTCCGCCAG
** *SMAD3* **	CAGGAGGAGAAGTGGTGCGA	TCCAGTGACCTGGGGATGGTAA
** *SMAD4* **	CTACCAGCACTGCCAACTTTCC	CCTGATGCTATCTGCAACAGTCC
** *SNAI1* **	TGCCCTCAAGATGCACATCCGA	GGGACAGGAGAAGGGCTTCTC
** *VE-CADHERIN 2* **	GGAGGAGCTCACTGTGGATT	CTGATGCAGCAAGGACAGC
** *VIM* **	GAGAACTTTGCCGTTGAAGC	GCTTCCTGTAGGTGGCAATC

## Data Availability

Data supporting the reported results can be found in our laboratory records. DNA, RNA, cell culture and tissue samples are part of the collection associated with the research of the group.

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
