# Peer review of "Endothelial-to-Mesenchymal Transition in an Hereditary Hemorrhagic Telangiectasia-like Pediatric Case of Multiple Pulmonary Arteriovenous Malformations"

_ijms, 2024, doi:10.3390/ijms25116163_

Round 1
Reviewer 1 Report
Comments and Suggestions for Authors
My major concern about this paper is the fact that the Authors state an endothelial to mesenchymal transition in cells obtained from a lung biopsy without any proof that the fibroblast-like cells observed in the patient-derived cell line are not the consequence of a "brought trough" of few fibroblasts that became the major cells in culture. No information is given on how the selection of endothelial cells was obtained starting from lung biopsy. Moreover, the article cited in the M&M section (ref#39)does not give additional information. On the contrary, it refers to another "manuscript in preparation" by Serrano-Heras that I was not able to find in PubMed. Moreover, in the cited article it is clearly demonstrated that the composition of cells derived with the used protocol is mixed and endothelial cells represent only a minority ((stromal cells (30–50 % CD99 + cells), endothelial cells (15–25 % CD34+ cells), and pericytes (30–40 % NG2+ cells)_ from ref#39).
In my opinion, a more detailed protocol for cell isolation and characterization before the culture is necessary, and a demonstration of the "purity" of the cell composition at the beginning of the culture is needed, to exclude a fibroblast-like cell initial "contamination".
In addition, I'm wondering why western blot analysis uses beta-actin as the reference protein, as the Authors stated an altered expression of this gene in the patient, when related to control. I think that this decision has to be explained both in M&M and in the results sections.
Another issue is in the last sentence of the first paragraph in the result section: "In spite of all the genetic analysis carried out in the known genes related to HHT and HHT-like syndromes, the genetic origin of the disease remains unknown." The paper, however, does not cite any genetic analysis performed.
So, at least a sentence on the genetic analysis method(s) used and the result(s) obtained should be added to the manuscript, and the negative results and the incomplete clinical picture of the child should be discussed.
This paper should represent an important finding of a novel disease-related mechanism so a piece of deeper information about these methods is, in my opinion, due.
Minor comments are included in the pdf of the paper I have attached.

I suggest revising the English language.
Author Response
Reviewer 1
My major concern about this paper is the fact that the Authors state an endothelial to mesenchymal transition in cells obtained from a lung biopsy without any proof that the fibroblast-like cells observed in the patient-derived cell line are not the consequence of a "brought trough" of few fibroblasts that became the major cells in culture. No information is given on how the selection of endothelial cells was obtained starting from lung biopsy. Moreover, the article cited in the M&M section (ref#39) does not give additional information. On the contrary, it refers to another "manuscript in preparation" by Serrano-Heras that I was not able to find in PubMed. Moreover, in the cited article it is clearly demonstrated that the composition of cells derived with the used protocol is mixed and endothelial cells represent only a minority (stromal cells (30–50 % CD99 + cells), endothelial cells (15–25 % CD34+ cells), and pericytes (30–40 % NG2+ cells) from ref#39).
Thank you very much for your concern that allows us a better clarification. We have deleted the reference that was not corresponding to the protocol of getting lung endothelial cells. We thank you for your detailed reading.
In my opinion, a more detailed protocol for cell isolation and characterization before the culture is necessary, and a demonstration of the "purity" of the cell composition at the beginning of the culture is needed, to exclude a fibroblast-like cell initial "contamination".
Thank you very much. To this end the following protocol has been added and highlighted in the Material and Methods section:
To isolate lung endothelial cells, mechanical deep disruption was made using scalpels to reduce pieces of around 30 mg to a homogenous mixture. EGM-2/EBM-2 basal medium (3 ml) (PromoCell, Heidelberg, Germany) without serum was added to the lung mixture and subjected to collagenase digestion for 20 minutes at 37ºC with gentle shaking in 5 ml sterile tubes. Then, 20 ml of EGM-2/EBM-2 basal medium (PromoCell) was added, and the mixture was filtered through a 70 µm cell strainer. The strainer was washed with additional 20 ml basal medium, and the flow-through was centrifuged at 1,250 rpm for 5 minutes. Supernatant was removed and pellet was washed with 20 ml Hanks Buffered Solution (HBSS) and centrifuged for additional 5 minutes. The resulting pellet was resuspended in 2 ml of EGM-2/EBM-2 endothelial growth complete medium containing all the specific endothelial supplements (Promocell), and 20% FBS plus anti mycotic-antibiotic solution (Gibco, New York, USA) and Mycozap as anti-mycoplasm. Cells were plated into 6 wells-plate previously coated with collagen type I. In the first week of culture, the medium was changed daily and replaced with fresh EGM-2/EBM-2 complete medium (PromoCell). At first days many cells from blood origin, were visible onto the plates. However, after 10-15 days of culture, clusters of cobblestone shaped cells became visible. EGM-2/EBM-2 complete medium (PromoCell) was replaced every day, from the 15th to the 30th day of culture, no other type was visible in the 6 wells-plate, except the cobblestone shaped cells, which became confluent after 30 days from the plating day. Cells from each confluent 6 wells-plate were detached by trypsin, passed to F-25 collagen coated flasks and cultured for 3 days with 5 ml of EGM-2/EBM-2 complete medium (PromoCell) (P-1 passage) until confluence. Some of the cells were frozen at this stage, (P-1 passage) at a density around 1 million cells. Some P-1 flasks were expanded further to 5 F-25 ml each, and frozen when flasks became confluent at passage 2 (1 million cells each).
Staining with an endothelial specific marker, such as von Willebrand factor, was performed to assure their endothelial nature. Neither fibroblast or any other cell type were found. Here see below the vWF staining.
In addition, I'm wondering why western blot analysis uses β-Actin as the reference protein, as the Authors stated an altered expression of this gene in the patient, when related to control. I think that this decision has to be explained both in M&M and in the results sections.
Thank you very much again. The point is well taken. We carried out the western blot analysis of the TGF-β/SMAD pathway, before doing the immunofluorescence microscopy of actin. When we realized about the decrease in β-Actin, we thought of this fact, we thought, as you mention, that beta-actin was not a good loading reference for the patient. However, the difference in the almost absence of SMADs in the patient was so clear. Therefore, we decided it was not necessary to repeat with another loading control as tubulin. Tubulin was used as loading control in the other WB, using the same lysates, in control and patient samples.
Another issue is in the last sentence of the first paragraph in the result section: "In spite of all the genetic analysis carried out in the known genes related to HHT and HHT-like syndromes, the genetic origin of the disease remains unknown." The paper, however, does not cite any genetic analysis performed.
So, at least a sentence on the genetic analysis method(s) used and the result(s) obtained should be added to the manuscript, and the negative results and the incomplete clinical picture of the child should be discussed.
Thanks again. Although the genetic analysis performed were negative in the search of mutations, we should have mentioned which analysis were done. To that effect we have included and highlighted the following paragraph, more than a single sentence, in Material and Methods section:
Sanger sequencing analysis of both germ-line DNA (peripheral blood) and from DNA of lung tissue was performed. The analysis included the exons and intron-exon boundaries of the two more common HHT pathogenic genes: ENG, ACVRL1/ALK1. After getting negative results, we carried out a genomic analysis of the classical and other HHT-related genes with specific capture of ENG, ACVRL1/ALK1, MADH4/SMAD4, BMP9 and RASA1 chromosome regions (Illumina platform). The analysis was done in DNA from blood and lung.
To discard big genomic reorganizations, we performed an optical genome of DNA samples from blood and lung. Clinical exome was also done in germline DNA of the test sample, using samples from his mother and his sister as control. Clinical exome with more than 500 reads depth, was made from the lung DNA to discard possible mosaicism.
The incomplete clinical picture of the child should be discussed.
Thank you for your remark, we have rephrased the first paragraph of results in the following way.
“The Index case of this study is an 8-years-old child with clinical diagnosis of probable HHT without a familiar history of HHT. The case shows incomplete Curaçao criteria: epistaxis and multiple diffuse PAVMs in both lungs. The symptoms were present since he was 3 years old, including strong cyanosis, hypoxemia (abnormal low oxygen in blood). This phenotype was becoming more severe and non-compatible with life, due to the low oxygen saturation. In 2020, he was put on top of an emergency list for pediatric lung transplantation. In spite of all the genetic analysis carried out in the known genes related to HHT and HHT-like syndromes, the genetic origin of the disease remains unknown.”
This paper should represent an important finding of a novel disease-related mechanism so a piece of deeper information about these methods is, in my opinion, due.
Minor comments are included in the pdf of the paper I have attached.
Thank you for your detailed revision of the typos and comments included in the PDF.

Reviewer 2 Report
Comments and Suggestions for Authors
Reviewer suggestions
The present article is a case report of 8-year-old patients having severe phenotypes of multiple diffuse PAVMs. The patient has an HHT-like molecular pattern. Authors have performed molecular biology and cellular functional analysis of primary endothelial cells isolated from the explanted lung and observed a loss of functionality in lung endothelial tissue and stimulation of endothelial to mesenchymal transition.
Recommendation
1. The research report is original and scientifically sound.
2. The result mentioned in the main body corroborated with the figures.
3. References are specific to the topic and sufficient.
4. Methods are written well.
Scientific comments
1. Provide suitable citations for lines 33–37.
2. Briefly discuss the role of SMAD in PAVMs either in the result or discussion section.
3. Provide a suitable reference for lines 47–54.
4. Lines 74–76, mention the major role of the BMPR2 gene in PAVM.
5. Line 213, In both cases, the advancing front moved. What does it mean here?
6. Line 261, HHT patient 14. check and delete 14.
Typo errors
1. Delete the full stop from the title.
2. 4 is in the power of 10. write it correctly throughout the MS.
3. Check the IJMS Reference style and modified accordingly.
Author Response
Reviewer 2
Reviewer suggestions
The present article is a case report of 8-year-old patients having severe phenotypes of multiple diffuse PAVMs. The patient has an HHT-like molecular pattern. Authors have performed molecular biology and cellular functional analysis of primary endothelial cells isolated from the explanted lung and observed a loss of functionality in lung endothelial tissue and stimulation of endothelial to mesenchymal transition.
Recommendation
- The research report is original and scientifically sound.
- The result mentioned in the main body corroborated with the figures.
- References are specific to the topic and sufficient.
- Methods are written well.
Scientific comments
- Provide suitable citations for lines 33–37.
Thank you very much, we have chosen two references to illustrate the first paragraphs.
References:
Kroon, S.; Van Thor, M.C.J.; Vorselaars, V.M.M.; Hosman, A.E.; Swaans, M.J.; Snijder, R.J.; Mager, H.J.; Post, M.C. The use of echo density to quantify pulmonary right-to-left shunt in transthoracic contrast echocardiography. Eur Heart J Cardiovasc Imaging 2021, 22, 1190-1196.
Parra, J.A.; Bueno, J.; Zarauza, J.; Fariñas-Alvarez, C.; Cuesta, J.M.; Ortiz, P.; Zarrabeitia, R.; Pérez del Molino, A.; Bustamante, M.; Botella, L.M.; Delgado, M.T. Graded contrast echocardiography in pulmonary arteriovenous malformations. Eur Respir J 2010, 35, 1279-85.
- Briefly discuss the role of SMAD in PAVMs either in the result or discussion section.
The almost absence of Smad 4 expression in the patient´s endothelial cells with the notion that SMAD4 is a central mediator of ENG, ACVRL1/ALK1 signaling, indispensable for the development of a proper arteriovenous network. In this case, lack of SMAD4 in the pulmonary ECs, plus extremely low levels of SMAD1/3 end in a completely altered pulmonary arteriovenous network.
Reference:
Kim, Y.H.; Choe, S.W.; Chae, M.Y.; Hong, S.; Oh, S.P. SMAD4 Deficiency Leads to Development of Arteriovenous Malformations in Neonatal and Adult Mice. J Am Heart Assoc 2018, 7, e009514.
- Provide a suitable reference for lines 47–54.
Al-Samkari, H. Hereditary hemorrhagic telangiectasia: systemic therapies, guidelines, and an evolving standard of care. Blood 2021, 137, 888-895.
- Lines 74–76, mention the major role of the BMPR2 gene in PAVM.
BMPR2 and the HHT genes (ACVRL1, ENG, and SMAD4) are all part of the transforming growth factor-beta signaling pathway. Pathogenic mutations in BMPR2 are associated mainly with hPAH.
Reference:
Machado, R.D.; Aldred, M.A.; James, V. et al. Mutations of the TGF-β type II receptorBMPR2 in pulmonary arterial hypertension. Hum Mutat 2006, 27, 121-132.
Several descriptions of patients with both PAVMs and PAH with pathogenic BMPR2 mutations have been also described.
References:
Soon, E.; Southwood, M.; Sheares, K.; Pepke-Zaba, J.; Morrell, N.W. Better off blue: BMPR-2 mutation, arteriovenous malformation, and pulmonary arterial hypertension. Am J Respir Crit Care Med 2014, 189, 1435-1436.
Scarpato, B.M.; McDonald, J.; Bayrak-Toydemir, P.; Elliott, C.G.; Cahill, B.C.; Emerson, L.L.; Keenan, L.M. The Shunt of It. Chest 2023, 163, e201-e205.
- Line 213, In both cases, the advancing front moved. What does it mean here?
In both types of cells, control and patient derived, there was migration after the disruption of the confluent layer. However, in the control, the advancing cells were moving in a coordinated way, as a front, while in the patient cells, these moved randomly not in an ordered manner.
- Line 261, HHT patient 14. check and delete 14. Done.
Typo errors
- Delete the full stop from the title. Done.
- 4 is in the power of 10. write it correctly throughout the MS. Done.
- Check the IJMS Reference style and modified accordingly. Done.
